# Factors Related to Caregiver Intentions to Vaccinate Their Children with Attention-Deficit/Hyperactivity Disorder against COVID-19 in Taiwan

**DOI:** 10.3390/vaccines9090983

**Published:** 2021-09-02

**Authors:** Ching-Shu Tsai, Ray C. Hsiao, Yu-Min Chen, Cheng-Fang Yen

**Affiliations:** 1Department of Child and Adolescent Psychiatry, Kaohsiung Chang Gung Memorial Hospital, Kaohsiung 833401, Taiwan; jingshu@cgmh.org.tw; 2College of Medicine, Chang Gung University, Taoyuan City 33302, Taiwan; 3Department of Psychiatry and Behavioral Sciences, University of Washington School of Medicine, Seattle, WA 98195, USA; rhsiao@u.washington.edu; 4Department of Psychiatry, Children’s Hospital and Regional Medical Center, Seattle, WA 98105, USA; 5Department of Psychiatry, Kaohsiung Medical University Hospital, Kaohsiung 80708, Taiwan; 6Department of Psychiatry, School of Medicine, College of Medicine, Kaohsiung Medical University, Kaohsiung 80708, Taiwan; 7College of Professional Studies, National Pingtung University of Science and Technology, Pingtung 91201, Taiwan

**Keywords:** attention-deficit/hyperactivity disorder, caregiver, COVID-19, vaccine

## Abstract

The aims of this study were to examine the proportion of caregivers who were hesitant to vaccinate their children with attention-deficit/hyperactivity disorder (ADHD) against coronavirus disease 2019 (COVID-19) and the factors related to caregiver intentions to vaccinate their children against COVID-19. In total, 161 caregivers of children with ADHD were recruited in this study. The caregivers completed an online questionnaire to provide data regarding their intention to vaccinate their children against COVID-19, concerns about the effectiveness and safety of vaccines, unfavorable family attitudes toward vaccines, and children’s medication use for ADHD and comorbid psychopathology. The factors related to caregiver intentions to vaccinate their child were examined using linear regression analysis. The results indicated that 25.5% of caregivers were hesitant to vaccinate their children with ADHD, and 11.8% refused to vaccinate their children against COVID-19. The caregivers’ concerns about the safety of vaccines and children’s regular use of medication for ADHD were negatively associated with caregiver intentions to vaccinate, whereas the children’s comorbid conduct or oppositional defiant problems were positively associated with the caregiver intentions to vaccinate. An intervention that enhances caregiver intentions to vaccinate their children with ADHD against COVID-19 by addressing the related factors found in this study is warranted.

## 1. Introduction

### 1.1. Vaccination against COVID-2019 in Children and Adolescents

COVID-19 has had disastrous consequences worldwide. As of 16 July 2021, the World Health Organization (WHO) has recorded 188,655,968 confirmed cases of COVID-19, including 4,067,517 deaths [1]. Vaccines are expected to stop the spread of COVID-19 [2]. Several vaccines are currently approved by the WHO for emergency use against COVID-19 [3,4]. On the basis of all available evidence, the WHO recommends vaccination for adults aged ≥18 years [5]. Review studies have concluded that despite the lower infection rate in children compared to adults, children remain susceptible to COVID-19 [6,7]; those with comorbidities are especially susceptible [8]. Therefore, the Centers for Disease Control and Prevention in the United States and the European Medicines Agency in the European Union have approved the Pfizer–BioNTech COVID-19 vaccine for use in children and teens aged ≥12 years to protect them against COVID-19 [9,10]. Caregiver intentions to vaccinate their children against COVID-19 becomes an important issue in the trend toward decreasing the age of vaccination against COVID-19.

### 1.2. Attention-Deficit/Hyperactivity Disorder as a Risk Factor for COVID-19

Attention-deficit/hyperactivity disorder (ADHD) has been found to be a risk factor for COVID-19 and poor outcomes. A nationwide database of adult patients in the United States revealed that those with ADHD had a significantly increased risk of COVID-19 [11]. Research in Israel found that untreated ADHD is a risk factor for COVID-19 infection and that drug treatment ameliorates this effect [12]; furthermore, ADHD was associated with the increased severity of COVID-19 symptoms and increased referral for hospitalization [13]. Another study among adults in Israel also found that ADHD symptoms are negatively associated with adaptation to the COVID-19 outbreak [14]. Previous study results have indicated that vaccination against COVID-19 is necessary to prevent COVID-19 infection and its adverse consequences in individuals with ADHD. To the best our knowledge, no study has examined caregiver intentions to vaccinate their children with ADHD against COVID-19 and related factors.

### 1.3. Factors Related to Caregiver Intentions to Vaccinate Their Children against COVID-19

Research has identified multiple factors related to the level of caregiver hesitancy to vaccinate their children against COVID-19 [15,16,17,18,19,20,21,22]. According to the theory of planned behavior [23], the factors related to caregivers’ vaccine hesitancy can be classified into three dimensions. The first is the caregivers’ attitudes toward vaccines; for example, caregivers worrying about the safety and effectiveness of COVID-19 vaccines [15,17,20,21] and low confidence in one’s knowledge about COVID-19 vaccines [16], are related to high hesitancy to vaccinate their children. The second is the caregivers’ perceived subjective norms regarding vaccination against COVID-19; for example, perceived support from family members in vaccinating their children against COVID-19 vaccination is related to low hesitancy [22]. The third is the caregivers’ perceived behavioral control; for example, perceived high self-confidence to successfully carry out vaccinating their children against COVID-19 is related to low hesitancy in caregivers [22]. In addition, according to the protection motivation theory (PMT) [24], threat appraisal and coping appraisal are two major cognitive processes that determine individual motivation to adopt protective behaviors to reduce the risk of contracting respiratory infectious diseases (RIDs) [25,26]. Regarding threat appraisal, research has found that caregiver concerns regarding COVID-19 infection is related to low hesitancy to vaccinate their children [20,21]. Regarding coping appraisal, low confidence in the effectiveness of COVID-19 vaccines is related to hesitancy to vaccinate their child [15,17,20,21]. The child being a younger age [16,18], a caregiver’s lower education level [17,18,19], a caregiver being of the female sex [18], and a caregiver’s exposure to negative information related to COVID-19 vaccination [22] are also related to high hesitancy for caregivers to vaccinate their children.

### 1.4. Issues Warranting Further Study in Children with ADHD

Several issues related to a caregiver’s intention to vaccinate their child with ADHD against COVID-19 warrant further study. First, factors related to a caregiver’s intention to vaccinate their child should be examined with consideration for ecological aspects [27]. The interaction between caregivers and their children with ADHD may differ from that between caregivers and their children without ADHD due to public and affiliated stigma toward ADHD and the care burden due to the behavioral problems in ADHD [28,29,30]. Further study is needed to examine whether factors related to a caregiver’s intention to vaccinate their child that have been identified in previous studies can be applied to caregivers of children with ADHD. Furthermore, no study has examined the effects of ADHD symptoms, comorbidity, or medication use on a caregiver’s intention to vaccinate their child with ADHD. According to PMT, self-efficacy in carrying out protective behaviors is a component of the cognitive appraisals that determine an individuals’ motivation to adopt protective behaviors to reduce the risk of RIDs [25,26]. Inattention may place individuals with ADHD at higher risk of forgetting to wear face masks or maintain social distancing [11,14]. Drug treatment for ADHD ameliorates the effect of ADHD symptoms on increasing the risk of COVID-19 infection [12]. Moreover, comorbid internalizing (e.g., depression) and externalizing (e.g., oppositional and conduct problems) problems in children with ADHD may increase their difficulties in cooperating with the reminders of their caregivers to adopt self-protective behaviors. In particular, several studies have found that emotional and behavioral problems worsened during the COVID-19 pandemic among children with ADHD [31,32,33]. Therefore, ADHD symptoms and emotional and behavioral comorbidities in children with ADHD may influence their ability to continue adopting self-protective behaviors against COVID-19 and increase caregiver intentions to vaccinate their children. However, the relationship between ADHD symptoms and comorbidity and caregiver intentions to vaccinate their children warrants further study.

### 1.5. Study Aims

The aims of this study were to examine factors related to caregiver intentions to vaccinate their children with ADHD against COVID-19. We hypothesized that caregiver factors (demographics, concerns about the effectiveness and safety of vaccines, and unfavorable family attitudes toward vaccines) and child factors (demographics, medication use for ADHD, worsening ADHD symptoms and depression during the COVID pandemic, and comorbid conduct and oppositional defiant problems) would be significantly related to caregiver intentions to vaccinate their children with ADHD against COVID-19.

## 2. Methods

### 2.1. Participants

This online questionnaire survey study was conducted between 13 October 2020 and 12 May 2021. Three representative associations for caregivers of children with ADHD in Taiwan agreed to post the link to our online questionnaire in their Facebook groups and in LINE (a direct messaging app) for caregivers of children with ADHD. Those who were interested in participating in this study could approach the online survey questionnaire via the link. The association A is headquartered in northern Taiwan and has several divisions in other areas of Taiwan; there were about 15,000 members in its Facebook and LINE groups at the time of this study. The association B is headquartered in middle Taiwan; there were about 7000 members in its Facebook and LINE groups. The association C is headquartered in southern Taiwan; there were about 1500 members in its Facebook and LINE groups. Anyone who is concerned with ADHD issues can join the Facebook and LINE groups of these three associations. Therefore, the members of the Facebook and LINE groups are not necessarily caregivers of children with ADHD. The members of the Facebook and LINE groups are also likely to overlap among these three associations.

Google Forms was used to host an online survey to collect data from the caregivers. At the beginning of the online questionnaire, we explained the goal, recruitment criteria, and procedures of this study to the potential respondents. The recruitment criteria were caregivers who were aged ≥20 years, who cared for children with ADHD, and who lived in Taiwan. Caregivers of children with ADHD could click the button “Agree to participate” to be redirected to the research questionnaire website or click the button “Decline to participate” to leave the website. The respondents were asked to answer all of the items on the questionnaires. We emphasized the anonymity and confidentiality of the online questionnaire and invited the caregivers of children with ADHD to participate in this study. Regarding to the sample size, we used the rule-of-thumb proposed by Green (*N* = 50 + 8 × number of independent variables) [34] to estimate the number of participants needed for linear regression analysis. There were 14 independent variables in this study; therefore, we estimated the number of participants to be 162. In total, 161 caregivers agreed to participate in this study, and 8 respondents declined to participate. This study was approved by the Institutional Review Board of Kaohsiung Medical University Hospital (KMUHIRB-EXEMPT(I) 20200018). Our study participants were provided no incentive to participate.

### 2.2. Measures

#### 2.2.1. Caregiver Intentions to Vaccinate Their Children with ADHD against COVID-19

Caregiver intentions to vaccinate their children with ADHD against COVID-19 was assessed using two items. The first item was “When a COVID-19 vaccine becomes available, will you let your child be vaccinated?” The responses to this item were “definitely willing,” “if my doctor recommends it, I would let my child receive it,” “not sure,” and “definitely not willing.” The second item was “Please rate your current willingness to let your child receive a COVID-19 vaccine,” which was rated on a Likert scale from 1 (very low) to 10 (very high) [35]. The result of the analysis of variance revealed that caregivers who rated “definitely willing” on the first item had the highest level of intention on the second item (9.0 ± 1.3), followed by “receive if my doctor recommends” (6.9 ± 1.8), “not sure” (4.4 ± 1.3), and “definitely not willing” (1.3 ± 0.6) (*F* = 124.736, *p* < 0.001). A previous study found that individual intention to receive vaccines against COVID-19 on the second item was significant associated with their past influenza vaccination uptake behaviors and their cognitive appraisals of vaccines against COVID-19 [36]. The questions and scoring are listed in Appendix A.

#### 2.2.2. Caregivers’ Concerns about Vaccines

We used three items on the Drivers of COVID-19 Vaccination Acceptance Scale (DrVac-COVID19S) [37,38] to ask the respondents how important their concerns about the safety of vaccines, effectiveness of vaccines, and their unfavorable family attitudes toward child vaccination were to their decision to vaccinate their children against COVID-19. A previous study found that individuals who adopted preventive COVID-19 infection behaviors had a significantly higher intention on the DrVac-COVID19S than those who did not adopt such behaviors; the result supported the known-group validity of the DrVac-COVID19S [37]. Each item was rated on a 4-point scale from 0 (*not important at all*) to 3 (*very important*). The questions and scoring are listed in Appendix A.

#### 2.2.3. Children Receiving Medication for ADHD

One question was related to the frequency at which the participants’ children used prescribed medication for ADHD. The question was rated on a 4-point scale from 0 (*n**ever*), to 1 (*seldom*), 2 (*sometimes*), or 3 (*often*). The questions and scoring are listed in Appendix A. Caregivers who responded with 3 and those who responded with <3 were considered to have children who regularly used and never or not regularly used medication for ADHD, respectively.

#### 2.2.4. Changes in ADHD and Depressive Symptoms during the COVID Pandemic

Four questions were related to changes in their children’s symptoms of inattention, hyperactivity, impulsivity, and depression before and during the pandemic. Each item was rated from 0 (*improved*), to 1 (*no change*), 2 (*mildly worsened*), or 3 (*significantly worsened*). The questions and scoring are listed in Appendix A. Respondents who responded 0 or 1 and those who responded 2 or 3 were classified as having children who did not and who did have worsening symptoms, respectively.

#### 2.2.5. Comorbid Conduct or Oppositional Defiant Problems

Two questions were related to whether their children had significant conduct and oppositional defiant problems before the COVID-19 pandemic. The questions and scoring are listed in Appendix A. Respondents who responded “yes” to any question were classified as having children who had comorbid conduct or oppositional defiant problems.

#### 2.2.6. Sociodemographic Characteristics

The caregivers’ sex (0 = *male*; 1 = *female*), age, and education level (0 = *high school or below*; 1 = *college or above*) were collected. The children’s sex (0 = *male*; 1 = *female*), and age were collected.

### 2.3. Statistical Analysis

Data analysis was performed using SPSS 24.0 (SPSS Inc., Chicago, IL, USA). The caregiver intentions to vaccinate their children, caregiver factors (demographics and concerns about vaccination), and child factors (demographics, treatment, and psychopathology) were analyzed using percentages and means with standard deviations (SDs). The association between the caregivers’ level of intention to vaccinate their children for COVID-19 (dependent variable) and the caregiver and child factors (independent variables) were first examined using univariate linear regression analysis. The independent variables that were significantly associated with the caregiver intentions in univariate linear regression analysis were further selected into stepwise multivariate linear regression analysis to examine their association with the caregiver intentions. A two-tailed *p* value of <0.05 indicated statistical significance.

## 3. Results

The caregiver intentions to vaccinate their children, their demographics, and their concerns about vaccination as well as the children’s demographics, treatments, and psychopathology are shown in Table 1. Of the 161 caregivers, 139 were female and 22 were male; their mean age was 42.8 years (SD = 5.9 years); 62.1% had an education level of college or higher. Regarding their intention to vaccinate, 37 (23.0%) caregivers reported that they were definitely willing to vaccinate, 64 (39.8%) reported that they were willing if their doctors recommended it, 41 (25.5%) were not sure, and 19 (11.8%) reported that they were definitely not willing to vaccinate. Of the 161 children, 131 were male and 30 were female; their mean age was 11.3 years (SD = 3.8 years); 70.2% used medication for ADHD regularly; 36.6% had comorbid conduct or oppositional defiant problems; 22.4–25.5% had worsened inattention, impulsivity, and depressive symptoms during the COVID-19 pandemic; and 11.8% had worsened hyperactivity.

The results of the univariate and stepwise multivariate linear regression analyses examining the association between the caregiver intentions to vaccinate their children and the caregiver and child factors are shown in Table 2. In the univariate linear regression analysis, the caregivers’ concerns about vaccine safety and effectiveness, unfavorable family attitudes toward vaccinating their children, and children’s regular use of medication for ADHD were negatively associated with the caregiver intentions to vaccinate, whereas comorbid conduct or oppositional defiant problems and worsened depression were positively associated with the caregiver intentions to vaccinate. Demographics and children’s worsened ADHD were not significantly associated with the caregiver intentions to vaccinate.

The caregivers’ concerns about vaccine safety and effectiveness, unfavorable family attitudes toward vaccinating their children, children’s regular use of medication for ADHD, comorbid conduct or oppositional defiant problems, and worsened depression were further selected into the stepwise multivariate linear regression analysis. The result of the stepwise multivariate linear regression analysis indicated that the caregivers’ concerns about the safety of vaccines and their children’s regular use of medication for ADHD were negatively associated with the caregiver intentions to vaccinate, whereas comorbid conduct or oppositional defiant problems were positively associated with the caregiver intentions to vaccinate. The interactions between the caregivers’ concerns about the safety of vaccines, their children’s regular use of medication for ADHD, and comorbid conduct or oppositional defiant problems were further selected into the stepwise multivariate linear regression analysis with enter method. The results indicated that the association of the three interaction variables with the caregiver intentions were not significant.

## 4. Discussion

The present study demonstrated that 25.5% of caregivers were hesitant to vaccinate their children with ADHD, and 11.8% refused to vaccinate their child against COVID-19. The caregivers’ concerns about their unfavorable family attitudes toward vaccination, children taking medication for ADHD, and their children’s comorbid conduct or oppositional defiant symptoms were significantly related to the caregivers’ level of intention to vaccinate their children against COVID-19.

### 4.1. Vaccine Hesitancy and Refusal in Caregivers of Children with ADHD

The present study demonstrated that 23.0% of the participants were definitely willing to vaccinate their child against COVID-19, whereas 37.3% of participants were hesitant (25.5%) or refused to vaccinate their children (11.8%). Given that no vaccine against COVID-19 was available in Taiwan during the study period [39], the participants might not have had sufficient knowledge to change their attitude toward vaccinating their children with ADHD. Nevertheless, the low rate of caregivers reporting definite willingness to vaccinate their child with ADHD in this study indicates that a program to enhance caregiver intentions to vaccinate their children with ADHD must be developed when vaccines become available for use in children. Nearly 40% of participants agreed to vaccinate their child if their doctors recommended it, indicating that recommendations from trustworthy doctors have a key role in convincing caregivers of the importance of vaccination against COVID-19 for children with ADHD.

### 4.2. Caregiver Concern about Vaccines

The results of our univariate linear regression analysis reveal that caregiver concern about vaccine safety and effectiveness and unfavorable family attitudes toward vaccination were negatively associated with caregiver intentions to vaccinate their children. Previous studies have found that concerns about the safety and effectiveness of COVID-19 vaccines among caregivers are important factors related to hesitancy to vaccinate their children [15,17,20,21]. Unlike vaccines for children that are developed over a long period and gain the trust of parents after years of use, the COVID-19 vaccines were developed in a very short period; people may be unsure of the efficacy of these vaccines in mitigating the harm posed by COVID-19, especially in children. Research has found that highlights that the novelty of these vaccines is one of the most common reasons reported by caregivers who refuse vaccination for their child [16,17].

The results of our multivariate linear regression analysis demonstrate that only caregiver concern about their family’s unfavorable attitudes but not concern about vaccine safety and effectiveness were significantly related to caregiver intentions. According to ecological systems theory [40], caring for children with ADHD is an issue of interaction among caregivers, children with ADHD, and other family members. Research has found that family support may reduce psychological stress and social stigma for caregivers of children with ADHD [41,42], whereas low perceived family support was associated with increased stress [43]. The attitudes that family members have towards vaccinating children against COVID-19 may directly hamper opportunities for caregivers to allow their children to receive the vaccine. The result of this study indicated the influence of family member attitudes towards child vaccination is that programs for enhancing knowledge and positive attitudes towards the vaccination of children against COVID-19 should focus not only on caregivers but also on the general population.

### 4.3. Medication for Treating ADHD and Comorbidity

The results of this study demonstrate that the regular use of medication for ADHD by children without worsened ADHD symptoms during the pandemic was negatively related with their caregiver intentions to vaccinate their children against COVID-19. A possible explanation for this negative relationship is that medication not only ameliorates ADHD symptoms but also the severity of rule-breaking behaviors [44,45] and thus increases cooperation with self-protection rules among children with ADHD. Caregiver concern about COVID-19 infection in their children may decrease, and they may perceive a low necessity to vaccinate their child. However, compliance with the recommendations for self-protective behavior does not guarantee immunity from COVID-19 infection. Mental health professionals should emphasize the importance of vaccination against COVID-19 in children with ADHD to caregivers in clinics.

Children’s comorbid conduct or oppositional defiant problems and worsened depressive symptoms during the pandemic were positively related to caregiver intentions to vaccinate their children in univariate linear regression analysis; however, only comorbid conduct or oppositional defiant problems remained significant. Conduct and oppositional defiant problems are prevalent in children with ADHD and can compromise the prognosis of ADHD [46]. Children with conduct or oppositional defiant disorder often argue with people in authority and actively refuse to comply with adults’ requests. This defiance inevitably increases the difficulties that caregivers face in requiring their child to follow the rules of self-protection during the pandemic. Caregivers may expect vaccination against COVID-19 to protect their children with conduct or oppositional defiant problems and ADHD from contracting COVID-19.

### 4.4. Implications

The present study is one of the first studies examining the level and related factors of caregiver intentions to vaccinate their children with ADHD. The present study revealed a high rate of caregivers with hesitancy to vaccinate their children with ADHD; trustworthy doctors have a key role in convincing caregivers of the importance of vaccination against COVID-19 for children with ADHD. The results may provide a reference for governments and health professionals in developing a program to enhance caregiver intentions to vaccinate their children with ADHD. A previous study on adults in Taiwan found that threat appraisal (including perceived vulnerability to and severity of COVID-19) and cognitive appraisal (including response and cost efficacy of vaccination, self-efficacy to have vaccination, and knowledge about vaccination) were the main factors influencing the intentions of adults to receive vaccination against COVID-19 [47]. Compared to the results of the study on adults, the present study demonstrated that caregiver concerns about their unfavorable family attitudes and their children regularly taking medication for ADHD treatment related to caregiver intentions to vaccinate their children with ADHD. The results remind health professionals that it is necessary to evaluate caregiver intentions to vaccinate their children in an ecological view; health professionals should take both children’s behaviors and families’ attitudes into consideration when developing programs for enhancing caregiver intentions to vaccinate their children with ADHD against COVID-19.

### 4.5. Limitations

This study has several limitations. First, the participants were enrolled through an online advertisement delivered to caregivers of children with ADHD who joined the associations for such caregivers in Taiwan. Although this was a practical method to recruit participants during the COVID-19 pandemic, this enrollment method may limit representative participants. Moreover, the sample size was small (*n* = 161); further study on a larger sample of caregivers of children with ADHD is warranted. To extend the range of participants, this study had no exclusion criteria and encouraged eligible caregivers of children with ADHD to participate. However, it also increased the heterogeneity of the participants. We had no information regarding the nonrespondents and could not determine the differences between the respondents and the nonrespondents. Second, the caregivers provided all of the data in this study, likely leading to the problem of shared-method variance resulting from a sole information source, which requires careful consideration. Moreover, whether there was social desirability bias in the data warrants further study; the validity of the caregiver responses to the items regarding their children receiving medication for ADHD, changes in ADHD and depressive symptoms, and comorbid conduct or oppositional defiant problems also warrants further study. Third, this study did not recruit the caregivers of children without ADHD for comparison. Further studies are warranted to examine whether the factors related to the intentions of the caregivers to vaccinate their children found in this study relate to those observed in caregivers of children without ADHD.

## 5. Conclusions

This study found that 37.3% of caregivers were hesitant or refused to vaccinate their children with ADHD against COVID-19. Furthermore, caregiver concerns about the safety of vaccines and their children’s use of medication for ADHD and comorbid conduct or oppositional defiant problems were related to caregiver intentions to vaccinate their children. On the basis of the results, we suggest that the caregivers who were hesitant or refuse to vaccinate their children require specific intervention programs to increase their intention in response to the trend toward decreasing the age of vaccination against COVID-19. Factors that related to caregiver intentions found in this study should be considered when developing the intervention programs.

## Figures and Tables

**Table 1 vaccines-09-00983-t001:** Caregiver intentions to vaccinate their children, demographics, and concern about vaccination and children’s demographics, treatment, and psychopathology (*N* = 161).

**Variables**	***n* (%)**	**Mean (SD)**	**Range**
*Caregivers*			
Caregiver intentions to vaccinate their child			
Category variable			
Definitely willing	37 (23.0)		
Willing if doctor recommend	64 (39.8)		
Not sure	41 (25.5)		
Definitely not willing	19 (11.8)		
Level of intention to vaccinate their child		6.1 (2.8)	1–10
Gender			
Male	22 (13.7)		
Female	139 (86.3)		
Age (years)		42.7 (5.9)	22–71
Education level			
Low (high school or below)	61 (37.9)		
High (college or above)	100 (62.1)		
Concern about the vaccines			
Safety		2.9 (0.5)	0–3
Effect		2.8 (0.5)	0–3
Unfavorable family attitude		1.6 (1.0)	0–3
*Children*			
Gender			
Boys	131 (81.4)		
Girls	30 (18.6)		
Age (years)		11.3 (3.8)	5–22
Taking medication for ADHD			
No or irregular	48 (29.8)		
Regular	113 (70.2)		
Comorbid conduct or oppositional defiant problems	59 (36.6)		
Having worsened psychopathology			
Inattention	40 (24.8)		
Hyperactivity	19 (11.8)		
Impulsivity	36 (22.4)		
Depression	41 (25.5)		

ADHD: attention-deficit/hyperactivity disorder.

**Table 2 vaccines-09-00983-t002:** Associations of caregiver and child factors related to caregivers’ intention to vaccinate their children against COVID-19: univariate and stepwise multivariate linear regression analysis.

Variables	Caregiver Intentions to Vaccinate Their Child against COVID-19		
	Univariate Linear Regression	Stepwise Multivariate Linear Regression	Enter Method of Multivariate Linear Regression
	B (SE)	B (SE)	B (SE)
*Caregivers*		-	
Female ^a^	−0.275 (0.655)	-	
Age	0.043 (0.038)		
High education level ^b^	−0.134 (0.464)	-	
Concern about vaccine safety	−1.583 (0.481) **	−1.260 (0.466) **	−1.591 (0.804)
Concern about vaccine effect	−1.109 (0.416) **		
Concern about unfavorable family attitude	−0.526 (0.220) *	-	
*Children*			
Girls ^c^	−0.632 (0.576)	-	
Age	0.031 (0.060)		
Regularly taking medication for ADHD ^d^	−1.399 (0.479) **	−1.175 (0.459) *	0.847 (2.870)
Comorbid conduct or oppositional defiant problems ^e^	1.558 (0.450) **	1.381 (0.435) **	−0.874 (2.823)
Worsened inattention ^f^	0.258 (0.520)	-	
Worsened hyperactivity ^g^	0.446 (0.697)	-	
Worsened impulsivity ^h^	0.647 (0.538)	-	
Worsened depression ^i^	1.037 (0.510) *	-	
*Interaction*			
Caregiver concerns about vaccine safety × Children regularly taking medication for ADHD			−0.602 (0.969)
Caregivers concerns about vaccines’ safety × Children’s comorbid conduct or oppositional defiant problems			0.949 (0.980)
Children regularly taking medication for ADHD × Children’s comorbid conduct or oppositional defiant problems			−0.699 (0.956)

ADHD: attention-deficit/deficit disorder; SE: standard error. ^a^: male caregivers as the reference; ^b^: high school or below as the reference; ^c^: male children as the reference; ^d^: not using or irregularly using medication as the reference; ^e^: no comorbid conduct or oppositional defiant problems as the reference; ^f^: no worsened inattention as the reference; ^g^: no worsened hyperactivity as the reference; ^h^: no worsened impulsivity as the reference; ^i^: no worsened depression as the reference; *: *p* < 0.05; **: *p* < 0.01.

## Data Availability

The data will be available upon reasonable request to the corresponding authors.

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
