# Peer review of "Factors Related to Caregiver Intentions to Vaccinate Their Children with Attention-Deficit/Hyperactivity Disorder against COVID-19 in Taiwan"

_vaccines, 2021, doi:10.3390/vaccines9090983_

Round 1

Reviewer 1 Report

please detail how the responders were recruited.

also add inclusion/exclusion criteria.

please specify how the questionnaire was administered.

please add the full questionnaire as supplementary material.

can you report the sample size estimation? 161 subjects seems a small sample size. Are you sure about the validity of your sample and then results?

Please add information about the validation of the questionnaire. Was it already validated? if yes, please cite the reference, if you developed it, please report also data on validation. 

authors stated that the questionnaire was an online questionnaire. please provide more details, which software was used, which channel, and so on.

How many potential respondents did you contact? What is the response rate? Do you have any information regarding non-respondents? Please add, if you do not have data, at least considerations and information on it.

In the discussion, can you add some comparison with previous research both in Taiwan in general, and specifically about ADHD caregivers and vaccine acceptance/refusal. In my view, now it is not exhaustive.

Can you add some consideration to the added value of your work? why was it important to conduct it? what are the public health implications of your results? 

In the limitation section, please also consider selection bias, as well as representativeness and social desirability bias. 

Author Response

We appreciated your valuable comments. As discussed below, we have revised our manuscript based on your comments. Please let us know if we need to provide anything else regarding this revision.

Comment 1

please detail how the responders were recruited.

Response

Thank you for your comment. We added the explanation for the method of recruiting the responders as below. Please refer to line 122-134.

Three representative associations for caregivers of children with ADHD in Taiwan agreed to post the link to our online questionnaire in their Facebook groups and in LINE (a direct messaging app) for caregivers of children with ADHD. Those who were interested in participating in this study could approach the online survey questionnaire via the link. The A association is headquartered in northern Taiwan and have several divisions in other areas of Taiwan; there was about 15,000 members in its Facebook and LINE groups at the time of this study. The B association is headquartered in middle Taiwan; there was about 7000 members in its Facebook and LINE groups. The C association is headquartered in southern Taiwan; there was about 1500 members in its Facebook and LINE groups. Anyone who are concerned with ADHD issues can join the Facebook and LINE groups of these three associations. Therefore, the members of the Facebook and LINE groups are not necessarily caregivers of children with ADHD. The members of the Facebook and LINE groups are also likely to overlap among these three associations.

Comment 2

also add inclusion/exclusion criteria.

Response

The inclusion criteria as below were described in section “2.1. Participants”. Please refer to line 137-138. To extend the range of participants, this study had no exclusion criteria and encouraged eligible caregivers of children with ADHD to participate in. However, it also increased the heterogeneity of the participants. We added it into the limitations as below to remind the readers as below. Please refer to line 350-352.

“The recruitment criteria were caregivers who were aged ≥20 years, cared for children with ADHD, and lived in Taiwan.”

To extend the range of participants, this study had no exclusion criteria and encouraged eligible caregivers of children with ADHD to participate in. However, it also increased the heterogeneity of the participants.

Comment 3

please specify how the questionnaire was administered.

Response

We added the explanation for the method of administrating the questionnaire as below. Please refer to line 124-126 and 139-141.

“Those who were interested in participating in this study could approach the online survey questionnaire via the link... Caregivers of children with ADHD could click the button “Agree to participate” to be redirected to the research questionnaire website or click the button “Decline to participate” to leave the website. The respondents were asked to answer all items on the questionnaires.”

Comment 4

please add the full questionnaire as supplementary material.

Response

We added the full questionnaire as supplementary material. Please refer to line 166, 176, 180, 188, and 194.

Comment 5

can you report the sample size estimation? 161 subjects seems a small sample size. Are you sure about the validity of your sample and then results?

Response

Thank you for your comment. We used the rule-of-thumb proposed by Green to estimate the number of participants needed for linear regression analysis. We added the explanation as below into the revised manuscript. Please refer to line 143-146. We agree that the total number of 161 participants was small. We listed it as one of the limitations in this study. Please refer to line 349-350.

“Regarding to the sample size, we use the rule-of-thumb proposed by Green (N = 50 + 8*number of independent variables) [34] to estimate the number of participants needed for linear regression analysis. There were 14 independent variables in this study; therefore, we estimated the number of participants to be 162.”

“Moreover, the sample size was small (N = 161); further study on a larger sample of caregivers of children with ADHD is warranted.”

Comment 6

Please add information about the validation of the questionnaire. Was it already validated? if yes, please cite the reference, if you developed it, please report also data on validation. 

Response

Thank you for your comment. We added the data of validation of questionnaires assessing the intention to vaccinate and concerns about vaccines as below into the revised manuscript. However, the validity of caregivers’ responses to the items for children’s receiving medication for ADHD, changes in ADHD and depressive symptoms, and comorbid conduct or oppositional defiant problems warrants further study. We added it into the limitations of this study.

2.2.1. Caregivers’ Intention to Vaccinate Their Children With ADHD Against COVID-19

“The result of analysis of variance revealed that caregivers who rated “definitely willing” on the first item had the highest level of intention on the second item (9.0±1.3), followed by “receive if my doctor recommends” (6.9±1.8), “not sure” (4.4±1.3), and “definitely not willing” (1.3 ±0.6) (F = 124.736, p < 0.001). A previous study found that individuals’ intention to uptake vaccines against COVID-19 on the second item was significant associated with their past influenza vaccination uptake behaviors and cognitive appraisals of vaccines against COVID-19 [36]. Please refer to line 159-165.

2.2.2. Caregivers’ Concerns About Vaccines

“We used three items on the Drivers of COVID-19 Vaccination Acceptance Scale (DrVac-COVID19S) [37,38] to ask the respondents how important their concerns about the safety of vaccines, effectiveness of vaccines, and their families’ unfavorable attitudes toward child vaccination were to their decision to vaccinate their children against COVID-19. A previous study found that individuals who adopted preventive COVID-19 infection behaviors had significant higher intention on the DrVac-COVID19S than those who did not adopt; the result supported known-group validity of the DrVac-COVID19S [37]. Please refer to line 172-174.

Limitations

“The validity of caregivers’ responses to the items for children’s receiving medication for ADHD, changes in ADHD and depressive symptoms, and comorbid conduct or oppositional defiant problems warrants further study. We added it into the limitations of this study.” Please refer to line 357-360.

Comment 7

authors stated that the questionnaire was an online questionnaire. please provide more details, which software was used, which channel, and so on.

Response

Thank you for your reminding. We added it as below into the revised manuscript. Please refer to line 135.

“Google Forms was used to host an online survey to collect data from the caregivers.”

Comment 8

How many potential respondents did you contact? What is the response rate? Do you have any information regarding non-respondents? Please add, if you do not have data, at least considerations and information on it.

Response

Thank you for your comment. We added the explanation for the potential respondents we contacted. Please refer to line 122-134. We did not have the information regarding non-respondents. We mentioned it as below into the limitations in the revised manuscript. Please refer to line 353-354.

Three representative associations for caregivers of children with ADHD in Taiwan agreed to post the link to our online questionnaire in their Facebook groups and in LINE (a direct messaging app) for caregivers of children with ADHD. Those who were interested in participating in this study could approach the online survey questionnaire via the link. The A association is headquartered in northern Taiwan and have several divisions in other areas of Taiwan; there was about 15,000 members in its Facebook and LINE groups at the time of this study. The B association is headquartered in middle Taiwan; there was about 7000 members in its Facebook and LINE groups. The C association is headquartered in southern Taiwan; there was about 1500 members in its Facebook and LINE groups. Anyone who are concerned with ADHD issues can join the Facebook and LINE groups of these three associations. Therefore, the members of the Facebook and LINE groups are not necessarily caregivers of children with ADHD. The members of the Facebook and LINE groups are also likely to overlap among these three associations.

We had no information regarding the nonrespondents and could not determine the difference between respondents and nonrespondents.

Comment 9

In the discussion, can you add some comparison with previous research both in Taiwan in general, and specifically about ADHD caregivers and vaccine acceptance/refusal. In my view, now it is not exhaustive.

Response

  1. Thank you for your suggestion. We added the comparison of the factors related to the intention to vaccinate against COVID-19 between the present study and a previous study on adults in Taiwan as below. Please refer to line 331-339.

“A previous study on adults in Taiwan found that threat appraisal (including perceived vulnerability to and severity of COVID-19) and cognitive appraisal (including response and cost efficacy of vaccination, self-efficacy to have vaccination, and knowledge about vaccination) were the main factors influencing adults’ intention to receive vaccination against COVID-19 [47]. Compared with the results of the study on adults, the present study demonstrated that caregivers’ concern about families’ unfavorable attitudes and children’s regularly taking medication for treating ADHD related to caregivers’ intention to vaccinate their children with ADHD.”

  1. To the best of our knowledge, there was no published study examining the caregivers’ intention to vaccinate their children with neurodevelopmental disorders. Therefore, we added the comparison with the results of the studies on caregivers’ intention to vaccinate their children with typical development as below. Please refer to line 279-287.

Previous studies have found that concerns about the safety and effectiveness of COVID-19 vaccines among caregivers are important factors related to hesitancy to vaccinate their children [15,17,20,21]. Unlike vaccines for children that are developed over a long period and gain parents’ trust after years of use, the COVID-19 vaccines were developed in a very short period; people may be unsure of the vaccines’ efficacy in mitigating the harm posed by COVID-19, especially in children. Research has found that the vaccines’ novelty is one of the most common reasons reported by caregivers to refuse vaccination for their child [16,17].

Comment 10

Can you add some consideration to the added value of your work? why was it important to conduct it? what are the public health implications of your results? 

Response

Thank you for your comment. We added a new paragraph “4.4. Implication” to describe the implication of the results as below. Please refer to line 324-343.

“4.4. Implication

The present study is one of the first studies examining the level and related factors of caregivers’ intention to vaccinate their children with ADHD. The present study revealed a high rate of caregivers with hesitancy to vaccinate their children with ADHD; trustworthy doctors have a key role in convincing caregivers of the importance of vaccination against COVID-19 for children with ADHD. The results may provide the reference for the governments and health professionals in developing the program to enhance caregivers’ intention to vaccinate their children with ADHD… The present study demonstrated that caregivers’ concern about families’ unfavorable attitudes and chidlren’s regularly taking medication for treating ADHD related to caregivers’ intention to vacciante their children with ADHD. The results reminded health professionals that it is necessary to evaluate caregivers’ intention to vacciante their chidlren in an ecological view; health professionals should take both children’s behaviors and families’ attitudes into consideration when developing the programs for enhancing caregivers’ intention to vaccinate their children with ADHDagainst COVID-19.”

Comment 11

In the limitation section, please also consider selection bias, as well as and social desirability bias. 

Response

Thank you for your comment. In the original manuscript, we had listed selection bias as the first limitation of this study. Please refer to line 345-349. In the revised manuscript, we also added the social desirability bias as another limitation. Please refer to line 357.

“…whether there was social desirability bias in the data warrants further study;...

Reviewer 2 Report

Title: Please mention "Taiwan" as study location.

l. 73: Can you give an example of "perceived behavioral control", like you did for the other two dimensions?

l. 122: What was the (approximate) size of the sampling frame, i.e. how many people could be theoretically reached via the three representative associations for caregivers of children with ADHD?

l. 185: Please mention that you used linear regression analysis.

l. 185: Did you use the AIC or the p-value (if so, which threshold) as criterion for the stepwise variable selection?

l. 185: Did you include interaction terms in the regression analysis?

Author Response

We appreciated your valuable comments. As discussed below, we have revised our manuscript based on reviewers’ suggestions. Please let us know if we need to provide anything else regarding this revision.

Comment 1

Title: Please mention "Taiwan" as study location.

Response

We added “in Taiwan” unto the title. Please refer to line 4.

Comment 2

  1. 73: Can you give an example of "perceived behavioral control", like you did for the other two dimensions?

Response

We added an example of "perceived behavioral control" as below into the revised manuscript. Please refer to line 72-75.

The third is caregivers’ perceived behavioral control; for example, perceived high self-confidence to successfully carry out vaccinating their children against COVID-19 is related to low hesitancy in caregivers [22].

Comment 3

  1. 122: What was the (approximate) size of the sampling frame, i.e. how many people could be theoretically reached via the three representative associations for caregivers of children with ADHD?

Response

Thank you for your comment. We added a paragraph to introduce the three associations for caregivers of children with ADHD and their members of their Facebook and LINE groups as below. We estimated that at least 15,000 individuals could be reached by this study, though not all individuals were caregivers of children with ADHD. Please refer to line 122-134.

“The A association is headquartered in northern Taiwan and have several divisions in other areas of Taiwan; there was about 15,000 members in its Facebook and LINE groups at the time of this study. The B association is headquartered in middle Taiwan; there was about 7000 members in its Facebook and LINE groups. The C association is headquartered in southern Taiwan; there was about 1500 members in its Facebook and LINE groups. Anyone who are concerned with ADHD issues can join the Facebook and LINE groups of these three associations. Therefore, the members of the Facebook and LINE groups are not necessarily caregivers of children with ADHD. The members of the Facebook and LINE groups are also likely to overlap among these three associations.”

Comment 4

  1. 185: Please mention that you used linear regression analysis.

Response

Thank you for your reminding. We added it into the revise manuscript. Please refer to line 207.

Comment 5

  1. 185: Did you use the AIC or the p-value (if so, which threshold) as criterion for the stepwise variable selection?

Response

We used the p-value (<0.05) as the criterion for the stepwise variable selection. We added the explanation as below into the revised manuscript. Please refer to line 207-211.

“The independent variables that were significant associated with caregivers’ intention in univariate linear regression analysis were further selected into stepwise multivariate linear regression analysis to examine their association with caregivers’ intention. A two-tailed p value of <0.05 indicated statistical significance.”

Comment 6

  1. 185: Did you include interaction terms in the regression analysis?

Response

We included the interaction terms in the regression analysis. The results indicated that the associations of the three interaction variables with caregivers’ intention were not significant. Please refer to line 245-249 and Table 2.

“The interactions between caregivers’ concern about the safety of vaccines, children’s regular use of medication for ADHD and comorbid conduct or oppositional defiant problems were further selected into the stepwise multivariate linear regression analysis with enter method. The results indicated that the associations of the three interaction variables with caregivers’ intention were not significant.”

Round 2

Reviewer 1 Report

Congratulatios. I'm satisfied with the changes provided.